# Nanochannel Array on Electrochemically Polarized Screen Printed Carbon Electrode for Rapid and Sensitive Electrochemical Determination of Clozapine in Human Whole Blood

**DOI:** 10.3390/molecules27092739

**Published:** 2022-04-24

**Authors:** Kai Wang, Luoxing Yang, Huili Huang, Ning Lv, Jiyang Liu, Youshi Liu

**Affiliations:** 1Key Laboratory of Integrated Oncology and Intelligent Medicine of Zhejiang Province, Department of Hepatobiliary and Pancreatic Surgery, Affiliated Hangzhou First People’s Hospital, Zhejiang University School of Medicine, Hangzhou 310006, China; kaiw3@zju.edu.cn; 2Key Laboratory of Surface & Interface Science of Polymer Materials of Zhejiang Province, Department of Chemistry, Zhejiang Sci-Tech University, Hangzhou 310018, China; 17858903877@163.com; 3Department of Psychiatry, Affiliated Xiaoshan Hospital, Hangzhou Normal University, Hangzhou 310018, China; hhl1626@163.com; 4Department of Pharmacy, The First Affiliated Hospital, School of Medicine, Zhejiang University, Hangzhou 310018, China; lvn1987@zju.edu.cn

**Keywords:** nanochannel array, screen-printed carbon electrode, electrochemical polarization, electrochemical determination of clozapine, human blood

## Abstract

Rapid and highly sensitive determination of clozapine (CLZ), a psychotropic drug for the treatment of refractory schizophrenia, in patients is of great significance to reduce the risk of disease recurrence. However, direct electroanalysis of CLZ in human whole blood remains a great challenge owing to the remarkable fouling that occurs in a complex matrix. In this work, a miniaturized, integrated, disposable electrochemical sensing platform based on the integration of nanochannel arrays on the surface of screen-printed carbon electrodes (SPCE) is demonstrated. The device achieves high determination sensitivity while also offering the electrode anti-fouling and anti-interference capabilities. To enhance the electrochemical performance of SPCE, simple electrochemical polarization including anodic oxidation and cathodic reduction is applied to pretreat SPCE. The electrochemically polarized SPCE (p-SPCE) exhibits an enhanced electrochemical peak signal toward CLZ compared with bare SPCE. An electrochemically assisted self-assembly method (EASA) is utilized to conveniently electrodeposit a vertically ordered mesoporous silica nanomembrane film (VMSF) on the p-SPCE, which could further enrich CLZ through electrostatic interactions. Owing to the dual signal amplification based on the p-SPCE and VMSF nanochannels, the developed VMSF/SPCE sensor enables determination of CLZ in the range from 50 nM to 20 μM with a low limit of detection (LOD) of 28 nM (S/N = 3). Combined with the excellent anti-fouling and anti-interference abilities of VMSF, direct and sensitive determination of CLZ in human blood is also achieved.

## 1. Introduction

Schizophrenia is a common mental illness that places a great burden on society. Some 20–30% of patients suffer from treatment-resistant schizophrenia (TRS) [1]. Research has shown that clozapine (CLZ) is the only drug with good therapeutic effect on TRS [2,3,4]. The intake of CLZ plays a crucial role in the recovery of patients. Excessive CLZ results in a lot of side effects including excessive salivation [5], agranulocytosis [6], seizures [7], weight gain [8], and mumps [9], which will cause secondary damage to the patient’s health [10]. Therefore, real-time determination of CLZ is of great significance to reduce the risk of relapse and readmission of patients.

Until now, the determination of clozapine has mainly relied on liquid chromatography-tandem mass spectrometry (LC-MS/MS), gas chromatography-mass spectrometry (GC-MS), colorimetric analysis and fluorescence sensing, etc. [11,12,13]. However, these detection strategies typically require expensive instruments and a professional operator. In addition, the pretreatment process for real sample analyses is often complex and time-consuming, and requires the use of a large amount of organic solvents [14,15,16,17,18,19]. Electrochemical methods have the characteristics of fast detection, simple instrumentation, high sensitivity, and convenient operation, and have been widely used in analyses of environmental, biological, medical, clinical or food samples [20,21,22,23,24,25,26,27,28]. Researchers have carried out electrochemical determination of CLZ using differential pulse voltammetry (DPV), square wave voltammetry (SWV), cyclic voltammetry (CV), etc. The determination sensitivity can be improved by introducing metal nanoparticles, carbon-based materials, etc. to the electrode surface [29,30,31]. However, the determination of CLZ in complex samples remains a great challenge. Complex biological or clinical samples (e.g., whole blood, serum, etc.) usually have complex matrices containing a large number of co-existing components including particulate matter (e.g., red blood cells, etc.), biological macromolecules (e.g., protein, DNA, etc.) and other electroactive small molecules (e.g., ascorbic acid-AA, uric acid-UA, etc.). On the one hand, particles or biological macromolecules will contaminate the surface of the electrode through non-specific adsorption, which will significantly reduce the detection sensitivity. On the other hand, co-existing electroactive small molecules can generate interfering signals. In addition, conventional electrochemical determination uses conventional electrodes (e.g., glassy carbon, gold, platinum, etc. with a diameter of 3 mm), so detection requires a larger sample volume. Therefore, it is very important to improve the anti-fouling, anti-interference ability and detection selectivity of the electrode while maintaining high detection sensitivity and low sample consumption.

The introduction of nanochannel arrays on the electrode surface is an effective strategy to improve its anti-fouling and anti-interference properties. In this regard, vertically ordered mesoporous silica nanomembrane films (VMSFs) have attracted much attention. VMSFs are composed of an array of silica nanochannels with uniform pore size (commonly 2–3 nm) and high porosity (up to 12 × 10^12^ cm^−2^). The high porosity ensures the mass transfer of small molecules within the film [32,33]. In addition, ultra-small nanochannels allow VMSF to achieve significant size exclusion for proteins or particles. On the other hand, the silanol groups (p*K*_a_ = 2–3) on the nanochannel surface endow VMSF with a certain charge selectivity [34,35,36], which can achieve selective permeation of charged species. In general, small molecules with negative charge will be repelled, making it difficult to for them enter the nanochannels and reach the electrode surface. In contrast, positively charged small molecules will be enriched, leading to signal amplification and high detection sensitivity [37,38,39]. Owing to the excellent anti-fouling [40] and anti-interference ability and potential enrichment effect, the VMSF modified electrode has great potential for direct and sensitive analyses of CLZ in complex samples.

Screen-printed electrodes (SPEs) are miniaturized and integrated electrodes prepared using screen-printing technology; they can disposable and entail low production costs. Since SPEs integrate the traditional three-electrode system (working electrode, counter electrode and reference electrode), the amount of sample used for detection can be very small. Among them, screen-printed carbon electrodes (SPCEs) have the advantages of a wide potential window, low background current, high biocompatibility and excellent chemical stability [41,42,43]. In this work, we constructed a miniaturized electrochemical sensing platform by equipping a SPCE with a nanochannel array which is able to directly and sensitively detect CLZ in human whole blood. In order to improve the electrochemical performance and achieve stable binding with VMSF, simple electrochemical polarization, including anodic oxidation at high potential and cathodic reduction at low potential, is applied to pretreat the SPCE. The electrochemically polarized SPCE (p-SPCE) exhibits a strong electrochemical peak signal in response to CLZ. An electrochemically assisted self-assembly method (EASA) is further utilized to conveniently electrodeposit VMSF on p-SPCE, which could further improve the electrochemical response to CLZ. Combined with the excellent anti-fouling and anti-interference abilities of VMSF, the constructed sensor can realize rapid and sensitive determination of CLZ in human blood. This work demonstrates a new strategy for the construction of a reliable, miniaturized, disposable and integrated electrochemical sensor with an anti-fouling layer and high detection sensitivity and selectivity.

## 2. Results and Discussion

### 2.1. Electrochemical Polarization of SPCE

As illustrated in Figure 1, SPCE was first treated by simple electrochemical polarization to enhance its electrochemical performance. Electrochemical polarization is a simple and green method to prepare highly active carbon electrodes. This method usually involves the electrochemical oxidization and reduction of electrodes in conventional electrolyte solutions. Usually, the electrode is first electrochemically oxidized at a high potential and then electrochemically reduced at a low potential. Thus, no complex chemical reagents and tedious operations are needed. During the polarization process, the sp^2^-conjugated carbon on the surface of the carbon electrode is oxidatively etched at high potential, resulting in abundant edge carbon, defects and oxygen-containing functional groups. These groups serve as electrocatalytic active sites which can not only enhance the adsorption of organic electroactive molecules, but also facilitate interfacial electron transfer reactions. In addition, electrochemical polarization facilitates the formation of porous surfaces. Therefore, electrochemically polarized carbon electrodes tend to exhibit high electrochemical response and significant electrocatalytic activity.

The cyclic voltammetry curves obtained in the electrochemical electrolyte or standard electrochemical probe solution (K_3_Fe(CN)_6_) on SPCE before and after electrochemical polarization are shown in Figure 2a. Compared with SPCE, p-SPCE showed a significantly increased charging current (Inset in Figure 2a), indicating an increase in the electroactive area of the electrode through electrochemical polarization. In the case of a redox probe, p-SPCE exhibited higher peak current and lower peak-to-peak separation, showing an improved electron transfer rate after electrochemical polarization. This phenomenon was further confirmed by electrochemical impedance spectroscopy (EIS), as shown in Figure 2b. It is well known that the charge transfer resistance (*R*_ct_) of an electrode is related to the diameter of the semicircular curve in the high frequency region. When the SPCE was electrochemically polarized, the p-SPCE displayed a reduced *R*_ct_ owing to the increased electron transfer rate.

### 2.2. VMSF Enquipment on the p-SPCE

After electrochemical polarization of the SPCE, VMSF was then electrodeposited on the electrode surface by the EASA method (Figure 1). EASA is a convenient method by which to apply VMSF to conductive substrates; it can complete the rapid growth of VMSF within 10 s. When the p-SPCE was immersed in an acidic precursor solution containing siloxane (TEOS) and cetyltrimethylammonium bromide (CTAB) micelles, a negative current was applied to the electrode and OH- was generated. This local pH increase induced the self-assembly of surfactant micelles (SM) and the polycondensation of siloxane, leading to a hexagonally packed pore structure. The stable binding of VMSF with p-SPCE was attributed to two factors. First, the negatively charged oxygen-containing functional groups on the surface of p-SPCE facilitated the electrostatic adsorption of the cationic SM. Second, the OH groups on p-SPCE formed Si-O bonds with VMSF through a co-condensation reaction, bestowing the adhesion of VMSF on p-SPCE with high mechanical stability. Finally, the SM in the nanochannels was removed with HCl-EtOH solution to obtain a VMSF/p-SPCE with open nanochannels.

Transmission electron microscopy (TEM) was used to characterize the morphology of the nanopore/channel structure of VMSF. Figure 3 demonstrates top-view and cross-sectional TEM images of the VMSF nanochannel. As shown in Figure 3a, the VMSF had a well-ordered structure with uniform pore size. The diameter of the nanopores was between 2 nm and 3 nm. The cross-sectional view in Figure 3b clearly reveals the parallel nanochannel structure.

The integrity of VMSF was further proven through electrochemical characterization. The CV curves and Nyquist plots of the obtained on SM@VMSF/p-SPCE and VMSF/p-SPCE are shown in Figure 3c,d. As seen, SM@VMSF/p-SPCE exhibited almost no peak current while VMSF/p-SPCE had remarkable peaks (Figure 3c). Thus, the filling of SM inside VMSF nanochannels inhibited the mass transfer of the redox probe through the nanochannels and then prohibited the subsequent electron transfer of between the electrode. EIS experiments also demonstrated the same phenomenon (Figure 3d). An electrode containing micelles in nanochannels showed extremely high charge transfer resistance. On the other hand, the VMSF/p-SPCE with open nanochannels presented significantly lower charge transfer resistance. The above results demonstrate the successful modification of VMSF on a p-SPCE and the integrity of the nanofilm.

### 2.3. Dual Signal Amplification and Significantly Enhanced CLZ Response on VMSF/p-SPCE

The CV and DPV curves of CLZ on different electrodes are compared in Figure 4. It can be seen that CLZ exhibited a pair of reversible redox peaks, which were attributed to its redox reaction on the electrode surface (Figure 4a). The response of bare SPCE to CLZ had the lowest peak current (Figure 4b). In contrast, the response of p-SPCE to CLZ was significantly enhanced. This was attributed to the enhancement of electrode performance by electrochemical polarization. On the one hand, electrochemical polarization increased the electroactive area, thereby increasing the interaction with the CLZ. On the other hand, the active groups generated by electrochemical polarization enhanced the electron transfer rate of p-SPCE. Therefore, electrochemical polarization can enhance the sensitivity to CLZ. When VSMF was grown on the surface of p-SPCE, VMSF/p-SPCE showed the highest peak current, demonstrating the enrichment of CLZ by VMSF nanochannels. This was attributed to the electron sieving effect of the nanochannels. In the electrolyte solution, the silanol groups on the surface of VMSF were negatively charged (p*K*_a_ = 2–3), which allowed electrostatic adsorption of CLZ (p*K*_a_ = 7.6) to occur. Thus, the VMSF/p-SPCE had a dual signal amplification effect on the determination of CLZ, indicating its potential for as a sensitive CLZ detector.

### 2.4. Optimization of Conditions for the Determination of CLZ

To improve the sensitivity to CLZ, conditions such as pH, enrichment time, and ionic strength were optimized. Figure 4c shows the DPV curves of CLZ on VMSF/p-SPCE at different pH. With the increase of pH, the oxidation peak potential of CLZ moved to the negative potential, and the peak current also changed greatly. The highest peak current was observed at pH 6. This was attributed to the effect of pH on the electrostatic interaction between the nanochannels with CLZ. An excessively low pH would reduce the negative charge on the surface of the VMSF nanochannels, which reduced the electrostatic force on CLZ. On the other hand, an overly high pH would reduce the positive charge of CLZ. Thus, pH = 6 was chosen for further investigation. Figure 4d displays the CV curves of CLZ at different scan rates. With the increase of scan rate, both the oxidation peak current and reduction peak current increased accordingly. Good linearity was observed between the peak current and the scan rate, indicating an adsorption-controlled process (inset in Figure 4d). Thus, the effect of enrichment time on the peak current of CLZ was investigate. As shown in Figure 4e, the peak current reached equilibrium after 60 s, indicating very short time for CLZ to reach mass transfer equilibrium on the electrode surface. This was attributed to the good permeability of VMSF and dual enrichment of the VMSF/p-SPCE. The ionic strength of the electrolyte solution affected the thickness of the electric double layer of the nanochannels, which changed the electrostatic interaction between the nanochannel and CLZ. Figure 4f shows the current responses of CLZ at different ionic strengths. When the ionic strength of the electrolyte solution was high, the electric double layer (EDL) did not overlap and the nanochannel exhibited electrostatic attraction to positively charged substances [44]. Thus, 0.1 M PBS was chosen as the electrolyte solution for the determination of CLZ.

### 2.5. Determination of CLZ Using VMSF/SPCE

The performance of the developed VMSF/p-SPCE sensor for the determination of CLZ using was investigated. Figure 5a displays the DPV curves obtained on the VMSF/p-SPCE in the presence of different concentrations of CLZ. As shown, the peak current gradually increased when the concentration of CLZ increased. A linear correlation was revealed between the oxidation peak current (*I*, μA) and concentration of CLZ (*C*, μM) in the range from 50 nM to 20 μM (*I* = 2.6 *C* − 0.05, *R*^2^ = 0.996) (Figure 5b). The limit of detection (LOD) was 28 nM with a signal-to-noise ratio of 3. For comparison, linear detection of CLZ ranging from 1 μM to 15 μM was obtained on a bare SPCE (*I* = 0.647 *C* + 0.052, *R*^2^ = 0.997). In the case of the p-SPCE, CLZ could be linearly detected from 0.5 to 20 μM (*I* = 1.44 *C* + 0.11, *R*^2^ = 0.993). Thus, VMSF/p-SPCE exhibited the highest sensitivity. The LOD obtained on the VMSF/p-SPCE was lower than that on GCE modified by multiwall carbon nanotubes or on WO_3_ nanoparticles hydride modified by α-terpineol [45], Bi–Sn nanoparticles applied to carbon aerogel-modified SPCE (Bi–Sn NP/CAG/SPCE) [46], ruthenium (IV) oxide nanoparticle-modified SPCE (RuO_2_NPs/SPE) [31] or a TiO_2_ nanoparticle-modified carbon paste electrode (TiO_2_NP-MCPE) [47]. The low LOD of the developed sensor may be attributed to the dual amplification effects resulting from both the p-SPCE and VMSF nanochannels.

### 2.6. Anti-Interference and Anti-Fouling Properties of the VMSF/p-SPCE Sensor

Anti-interference and anti-fouling capabilities are important characteristics of electrochemical sensors, and are especially important for analyses of real samples with complex matrices. To study the anti-interference ability of the developed VMSF/p-SPCE sensor, the effects of common substances in human blood on the determination of CLZ were investigated. Briefly, electrolyte ions (K^+^, Na^+^, Mg^2+^, Ca^2+^), common redox small molecules (ascorbic acid-AA, dopamine-DA, uric acid-UA) and metabolites (urea, glucose-Glu) were tested. As shown in Figure 6a, determinations of CLZ were not affected even when the concentrations of the above substances were 25 times higher than that of CLZ, proving the good anti-interference characteristic of the sensor. It is particularly noteworthy that three electroactive small molecules, i.e., AA, DA, and UA, did not interfere with the determination of CLZ, indicating that the constructed VMSF/p-SPCE sensor has excellent potential resolution. Blood samples or tablets associated with CLZ often contain large amounts proteins, hydroxypropyl methyl cellulose (HPMC) and starch. Therefore, the effects of bovine serum albumin (BSA), HPMC and soluble starch on the determination of CLZ were investigated. As shown in Figure 6b–d, these three species did not interfere with the determination of CLZ, further revealing the good anti-fouling ability of the electrode. In contrast, the responses changed significantly in presence or absence of these species (inset in Figure 6b–d). The excellent anti-fouling performance of the VMSF/p-SPCE sensor was attributed to the size and electrostatic exclusion effect of the VMSF nanochannels.

### 2.7. Determination of CLZ in Human Whole Blood with Low Sample Consumption

The excellent anti-fouling and anti-interference properties of the constructed VMSF/p-SPCE sensor show its great potential for use in direct electroanalyses of CLZ in complex samples. As a proof-of-concept, electrochemical determination of CLZ in human whole blood was investigated. Human whole blood was applied as the detection medium after dilution by a factor of 100. For comparison, determinations using a bare SPCE or p-SPCE were also undertaken. As shown in Figure 7a, the DPV peak current on the VMSF/p-SPCE increased with an increase in the concentration of CLZ. Although the detection sensitivity was reduced compared with that of the buffer system, the VMSF/p-SPCE sensor could still linearly detect CLZ from 0.5 μM to 12 μM (*I* = 0.563 *C* + 0.04, *R*^2^ = 0.996) with a LOD of 110 nM (Figure 7b). In contrast, the CLZ signal on both the bare SPCE and p-SPCE showed a non-linear relationship with its concentration, indicating that the electrode was seriously contaminated. The regeneration and reusability of the constructed VMSF/p-SPCE sensor was investigated. The performance of the p-SPCE was also examined for comparison. The sensor can easily be regenerated by soaking in a hydrochloric acid–ethanol solution (0.1 M) for 2 min. As shown in Figure 7c,d, both electrodes can be easily regenerated, although the CLZ signals on the regenerated electrodes were almost undetectable in the electrolyte. In addition, the VMSF/p-SPCE sensor can be reused with no significant change in its response to CLZ (Figure 7c). However, the peak currents for CLZ obtained on the p-SPCE were significantly reduced (Figure 7d). This phenomenon once again serves as evidence of the excellent anti-fouling performance of VMSF.

The reliability of the device was further investigated by the standard addition method. Different concentrations of CLZ were artificially added to the human whole blood. Then, the whole blood was diluted by a factor of 10. As shown, the recovery of the concentration of CLZ ranged from 97.3% to 104%, with a relative standard deviation (RSD) of no more than 3.2% (Table 1), indicating good reliability for direct measurements of CLZ in complex samples. Since the SPCE features an integrated working electrode, a counter electrode and a reference electrode, a very small amount of sample (50 μL) can be dropped directly onto the surface of the electrode during tests. Therefore, the constructed VMSF/p-SPCE sensor can achieve rapid, direct and sensitive determinations of CLZ with very low sample consumption, indicating its great potential for use with fingertip blood.

## 3. Materials and Methods

### 3.1. Chemicals and Materials

Tetraethyl orthosilicate (TEOS), hexadecyl trimethyl ammonium bromide (CTAB), potassium ferricyanide (K_3_[Fe(CN)_6_]), tetrapotassium hexacyanoferrate trihydrate (K_4_[Fe(CN)_6_]), sodium phosphate dibasic dodecahydrate (Na_2_HPO_4_•12H_2_O), potassium hydrogen phthalate (KHP), glucose (Glu), ascorbic acid (AA), uric acid (UA), dopamine hydrochloride (DA), bovine serum albumin (BSA), hydroxypropyl methylcellulose (HPMC), soluble starch (Starch), sodium dodecyl sulfate (SDS) and clozapine (CLZ) were purchased from Aladdin Biochemical Technology Co., Ltd. (Shanghai, China). Potassium chloride (KCl), anhydrous calcium chloride (CaCl_2_) and anhydrous ethanol (EtOH) were purchased from Hangzhou Gaojing Fine Chemical Co., Ltd. (Hangzhou, China). Magnesium chloride (MgCl_2_) and sodium dihydrogen phosphate dihydrate (Na_2_H_2_PO_4_•2H_2_O) were purchased from Shanghai Macklin Biochemical Technology Co., Ltd. (Shanghai, China). Sodium chloride (NaCl) and urea (Urea) were purchased from Tianjin Yongda Chemical Reagent Co., Ltd. (Tianjing, China). Screen-printed carbon electrodes (SPCEs) were purchased from Metrohm (Bern, Switzerland). Briefly, SPCEs contain three integrated electrodes, i.e., working and counter electrodes made up of conductive graphite paste and an Ag reference electrode comprising conductive silver paste. Human whole blood (healthy male) was provided by the Hangzhou Occupational Disease Prevention and Control Institute (Hangzhou, China). All reagents used in the experiment were of analytical grade and did not require further processing. The ultrawater (18 MΩ·cm) used in the experiments was prepared by the Mill-Q system (Millipore Corporation, Burlington, MA, USA).

### 3.2. Experiments and Instrumentations

The pore structure and vertically ordered nanochannels of the VMSF were characterized by transmission electron microscopy (TEM). Images were obtained on a transmission electron microscope (HT7700, Hitachi, Japan). The accelerating voltage was 100 kV. VMSF was gently scraped from p-SPCE, then dispersed in ethanol. Before characterization, the VMSF dispersion was dropped onto a copper grid. All electrochemical experiments, including cyclic voltammetry (CV), differential pulse voltammetry (DPV) and electrochemical impedance spectroscopy (EIS), were performed using an Autolab electrochemical workstation (PGSTAT302N, Metrohm, Herisau, Switzerland). The CV scan rate was 50 mV/s. During the DPV test, the step potential was 0.005 V, the pulse amplitude was 0.05 V, the pulse time was 0.05 s and the interval time was 0.2 s. EIS measurement was performed in K_3_Fe(CN)_6_/K_4_Fe(CN)_6_ (2.5 mM) solution containing KCl (0.1 M) with a frequency range of 10^4^ to 10^−1^ Hz.

### 3.3. Electrochemical Polarization of SPCE

Electrochemical polishing was performed to remove impurities such as organics and polymers from the surface of SPCE electrodes. Briefly, H_2_SO_4_ (0.05 M) was used as the medium and CV scans were then performed for 10 cycles between 0.4 and 1.0 V. After the electrode was rinsed with ultrapure water and dried with N_2_, it was placed in a phosphate buffer solution (PBS, 0.1 M, pH = 5). The subsequent electrochemical polarization included anodic oxidation and cathodic reduction. Firstly, the electrode was anodized by applying a constant potential of 1.8 V for 300 s. The oxidized SPCE was then cathodically reduced by CV scanning for three cycles (initial potential: 0 V; minimum potential: −1.3 V; maximum potential: 1.25 V; and scan rate: 100 mV/s). The obtained electrochemically polarized SPCE, that is, p-SPCE, was washed with ultrapure water and then dried with N_2_.

### 3.4. Preparation of VMSF-Modified p-SPCE

VMSF was electrodeposited on the surface of p-SPCE by the EASA method. To prepare the precursor solution for VMSF deposition, ethanol (20 mL) was mixed with an equal volume of NaNO_3_ (0.1 M, pH = 2.6). Then, CTAB (1.585 g) was added under magnetic stirring until the solution became clear. After TEOS (3050 μL) was added, the obtained solution was stirred at room temperature for 2.5 h. To grow VMSF, a p-SPCE electrode was placed in the precursor solution and a galvanostatic current (−52.2 μA) was applied for 10 s. As surfactant micelles (SM) were present within VMSF nanochannels, the obtained electrode was denoted as SM@VMSF/p-SPCE. After thorough rinsing with ultrapure water, the SM@VMSF/p-SPCE was aged at 80 °C for 10 h. Then, SM was removed by immersing the SM@VMSF/p-SPCE in 0.1 M HCl-EtOH solution (*v*:*v* = 1:1) for 5 min with stirring. An electrode with open nanochannels was finally obtained and named VMSF/p-SPCE.

### 3.5. Electrochemical Determination of CLZ

PBS (0.1 M, pH = 6) was used as the detection electrolyte. To investigate the effect of a complex medium on detection, human whole blood was also applied as the detection medium after diluting by a factor of 100 using PBS electrolyte. Electrochemical determination of clozapine was carried out by DPV. After adding different concentrations of CLZ, a DPV curve was recorded. For the real sample analysis, different concentrations of CLZ were added to the human whole blood to simulate changes in CLZ levels in patient blood after taking the drug. Then, human whole blood with added CLZ was diluted by a factor of 10 using PBS electrolyte. Next, 50 μL of the obtained solution was dropped on the surface of VMSF/p-SPCE electrode. After static enrichment for 1 min, a DPV curve was recorded.

## 4. Conclusions

In summary, using simple equipment, a miniaturized, integrated and disposable electrochemical sensing platform was constructed comprising a vertically ordered mesoporous silica nanomembrane film (VMSF) on electrochemically polarized SPCE (p-SPCE). The p-SPCE was characterized by increased active area and improved electron transfer rate, thereby improving the determination sensitivity of CLZ. The nanochannels of the VMSF also achieved electrostatic enrichment toward CLZ. Due to the excellent size/charge exclusion effect of the nanochannels and the high potential resolution ability of p-SPCE, the constructed VMSF/p-SPCE sensor had excellent anti-interference and anti-fouling properties, allowing it to realize rapid, direct and sensitive determinations of CLZ in human whole blood. In addition, the developed VMSF/p-SPCE sensor has the advantage of very low sample consumption (50 μL), indicating great potential for use with fingertip blood. The nanochannel array on SPCE established here also provides a new strategy for rapid, direct and sensitive electroanalyses of complex samples.

## Figures and Tables

**Figure 1 molecules-27-02739-f001:**
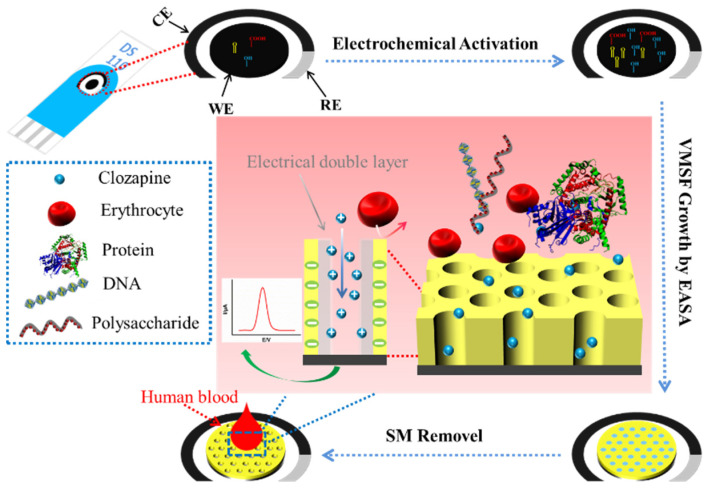
Schematic of the VMSF equipment on the electrochemically polarized SPCE and the direct determination of CLZ in human whole blood with high anti-fouling and anti-interference abilities.

**Figure 2 molecules-27-02739-f002:**
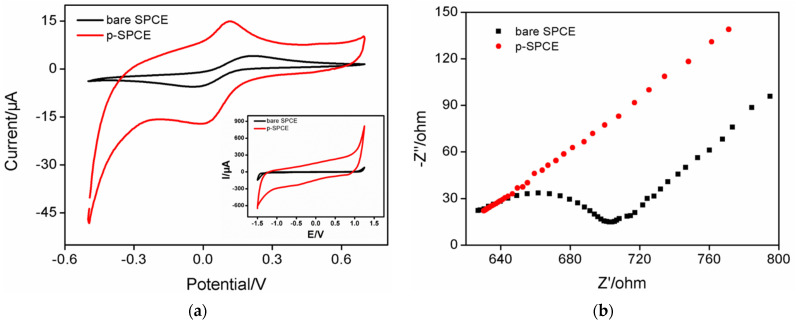
(**a**) CV curves obtained using different electrodes in 0.05 M KHP solution containing 0.5 mM Fe(CN)_6_^3+^. Inset: CV curves obtained with SPCE or p-SPCE in PBS (0.1 M, pH = 6). (**b**) Nyquist plots of different electrodes obtained in 0.1 M KCl solution containing 2.5 mM K_3_Fe(CN)_6_ and 2.5 mM K_4_Fe(CN)_6_.

**Figure 3 molecules-27-02739-f003:**
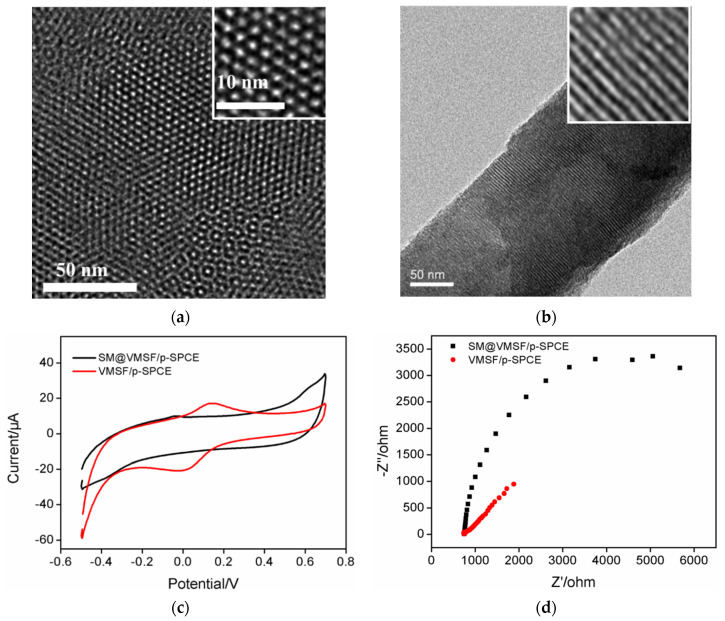
Top-view (**a**) and cross-sectional (**b**) TEM images of VMSF at different magnifications. (**c**) CV curves obtained at different electrodes in 0.05 M KHP solution containing 0.5 mM Fe(CN)_6_^3+^. (**d**) Nyquist plots of different electrodes obtained in 0.1 M KCl solution containing 2.5 mM K_3_Fe(CN)_6_ and 2.5 mM K_4_Fe(CN)_6_.

**Figure 4 molecules-27-02739-f004:**
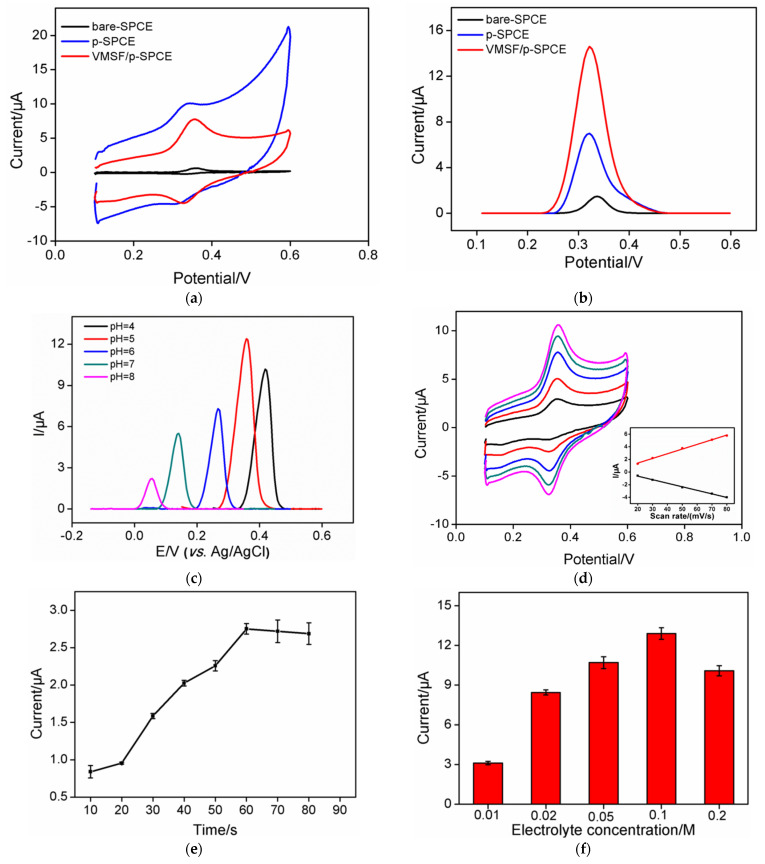
(**a**) CV or DPV (**b**) curves of CLZ on different electrodes. (**c**) DPV curves of CLZ at different pH. (**d**) CV curves of CLZ at different scan rates. The inset shows the linear regression curve between the peak current and the scan rate. (**e**) Effect of enrichment time on the peak current of CLZ. (**f**) Effect of the concentration of the electrolyte on the peak current of CLZ.

**Figure 5 molecules-27-02739-f005:**
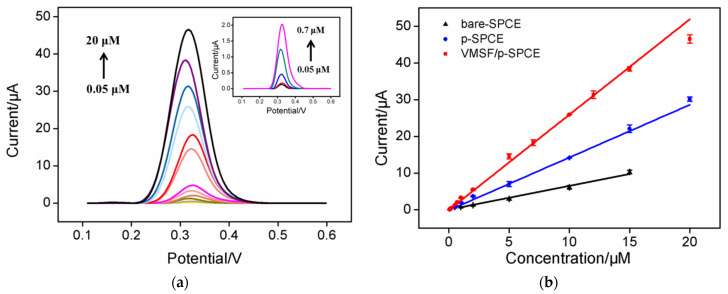
(**a**) Differential pulse voltametric curves of VMSF/p-SPCE toward various concentrations (0.05, 0.1, 0.2, 0.5, 0.7, 1, 2, 5, 7, 10, 12, 15, 20 μM) of CLZ. The inset is an enlarged image of the curves at low concentrations. (**b**) Corresponding calibration curves for the determination of CLZ using VMSF/p-SPCE, p-SPCE, or bare SPCE.

**Figure 6 molecules-27-02739-f006:**
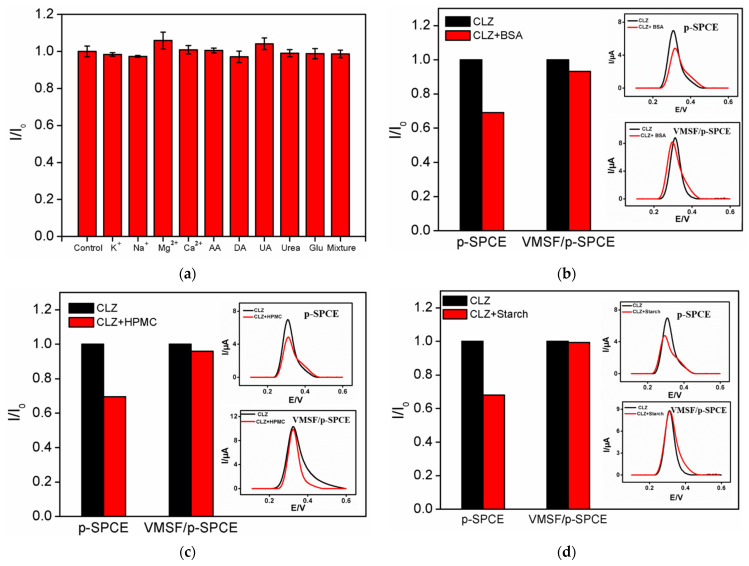
(**a**) The peak current ratio of CLZ on the VMSF/p-SPCE in the absence (*I*_0_) or presence (*I*) of the indicated substance or mixtures thereof. (**b**–**d**) The peak current ratio of CLZ on the VMSF/p-SPCE or p-SPCE in the absence (*I*_0_) or presence (*I*) of BSA (**b**), HPMC (**c**) and starch (**d**). Insets are the corresponding DPV curves obtained using single CLZ (black) or the binary mixture (red).

**Figure 7 molecules-27-02739-f007:**
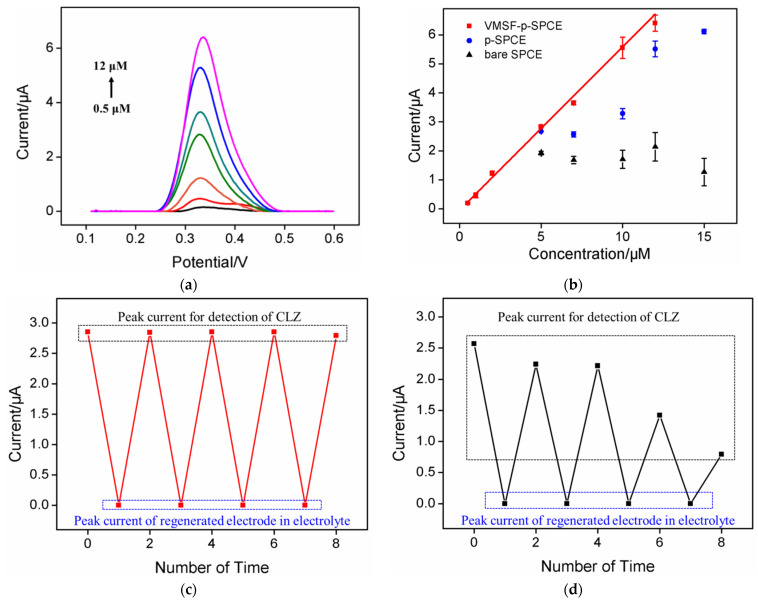
(**a**) DPV curves obtained on VMSF/p-SPCE in diluted human whole blood with the addition of different concentrations of CLZ (from bottom to top: 0.5, 1, 2, 5, 7, 10, 12 μM). (**b**) The corresponding calibration curve obtained on the VMSF/p-SPCE and DPV peak currents obtained on both the p-SPCE and bare SPCE. The regeneration and reuse performance of the VMSF/p-SPCE (**c**) and p-SPCE (**d**). The first peak current was obtained using the original electrode. Other peak currents were obtained in electrolyte (**bottom**) or CLZ solution (**top**) using the regenerated electrodes. The detected CLZ was 5 μM in diluted human blood (i.e., by a factor of 100).

**Table 1 molecules-27-02739-t001:** Determination of CLZ in human whole blood samples.

Sample ^a^	Added (μM)	Found (μM)	RSD (%)	Recovery (%)
Human whole blood ^a^	10.0	10.4	2.9	104
	30.0	29.2	3.2	97.3
	50.0	51.0	2.6	102

^a^ Samples with added CLZ were diluted 10 times using PBS electrolyte. The concentration of CLZ is before dilution.

## Data Availability

The data presented in this study are available on request from the corresponding author.

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
