# Peer review of "Nanochannel Array on Electrochemically Polarized Screen Printed Carbon Electrode for Rapid and Sensitive Electrochemical Determination of Clozapine in Human Whole Blood"

_molecules, 2022, doi:10.3390/molecules27092739_

Round 1

Reviewer 1 Report

This work reports a equipment of nanochannel array described  for the detection of clozapine. There are same aspects which were not clear for me and I would ask the authors make it explicit within a revision.

  1. How many times can the array be reused? It is said that it has a high antifouling capacity but it is not said how many times it can be used.
  2. Should be compared with other methods described in the literature
  3. I wonder if the authors could determine CLZ in real samples, since since they have to dilute 100 times, if the sample contains 1µM, after diluting you would have 0.01µM, below the LOD.
    The authors should first contaminate and then dilute in order to resemble a real sample. I think it is a very important concept that the authors should correct, and see if they can dilute it less, otherwise it could not be determined in real samples.

Author Response

Comment 1: “How many times can the array be reused? It is said that it has a high antifouling capacity but it is not said how many times it can be used.”

Answer: We thank the reviewer for the insightful comment. As suggested, the regeneration and reusability of the developed sensor have been investigated. The results have been shown as new Figure 7 c and d. In addition, the related discussion has also been added in the revision (See section 3.6. Detection of CLZ in Human Whole Blood with Low Sample Consumption).

Comment 2: “Should be compared with other methods described in the literature.”

Answer: We thank the reviewer for the helpful comment. As suggested, the performance of the developed sensor has been compared with other methods described in the literature. The related discussion and references have been added in the revision (See section 3.4 Sensitive Detection of CLZ Using VMSF/SPCE).

Comment 3: “I wonder if the authors could determine CLZ in real samples, since they have to dilute 100 times, if the sample contains 1µM, after diluting you would have 0.01µM, below the LOD. The authors should first contaminate and then dilute in order to resemble a real sample. I think it is a very important concept that the authors should correct, and see if they can dilute it less, otherwise it could not be determined in real samples.”

Answer: We thank the reviewer for the constructive comment. According, the human whole blood is detected with low dilution factor (by a factor of 10). Results is shown as new Table 1 and the related discussion has been added in the revision (See section 3.6. Detection of CLZ in Human Whole Blood with Low Sample Consumption). 

Reviewer 2 Report

A thoughtfully designed study. The authors use an integrated and synergistic method to solve a relevant clinical issue. The quality of the results and presentation is very high.

A few minor remarks: 

1) Electrochemical polishing - is the carbon material being dissolved into a solution during the procedure? Or is it just an electrochemical pre-treatment? 

2) The electrogenerated OH radicals are typically generated only in neutral or high pHs

3) Anodized electrodes typically have a lower charging current as they become hydrophilic. The rising charging current results from a bigger active surface or the presence of reduced electrochemically active groups that were introduced during the cathodization of the electrode.

4) The silanol group pKa 2-3 is relatively low. Please provide a reference.

5)  95 % of Clozapine binds to the blood plasma proteins. (According to Ereshefsky L, Watanabe MD, Tran-Johnson TK. Clozapine: an atypical antipsychotic agent. Clin  Pharm 1989;8:691-709). This could pose a problem for very strict (e.g., FDA) verification procedures.

6) Please provide curves at the concentration levels around the LOD. The lowest presented concentrations are at concentrations two orders higher.

The Language is simple, descriptive, and understandable with minimum grammar errors.

7) "To improve the sensitivity for the detecting CLZ, the detection conditions such as pH, enrichment time, and ionic strength are optimized." - use past tense (were optimized)

8) Several terms can be improved to convey the correct meaning.
E.g.:
detection (analyte present/not present) vs. determination (what is the concentration of the analyte)

9) Measurement vs. experiment

10) "fouling" generally means getting a decreased signal after repeated experiments at a single electrode, typically by covering the electrode with polymers etc. Anti-fouling properties in this sense are not really necessary for a disposable sensor.

Author Response

General comment: “A thoughtfully designed study. The authors use an integrated and synergistic method to solve a relevant clinical issue. The quality of the results and presentation is very high.”

Answer: We thank the reviewer for the positive and constructive comments. Accordingly, revision with new experimental data, more discussion and error correction has been made. We hope that the reviewer would now find this revision acceptable for publication.

Comment 1:“Electrochemical polishing - is the carbon material being dissolved into a solution during the procedure? Or is it just an electrochemical pre-treatment?”

Answer: We thank the reviewer for the comment. Electrochemical polishing is used to remove impurities such as organics and polymers from the surface of SPCE electrodes. This sentence has been added in the revision for clear description. (See section 2.3. Electrochemical Polarization of SPCE)

Comment 2:“The electrogenerated OH radicals are typically generated only in neutral or high pHs.”

Answer: We thank the reviewer for the helpful comment. As suggested, the description in the revision has been corrected (See section 3.1. Electrochemical Polarization of SPCE).

Comment 3:Anodized electrodes typically have a lower charging current as they become hydrophilic. The rising charging current results from a bigger active surface or the presence of reduced electrochemically active groups that were introduced during the cathodization of the electrode.

Answer: We thank the reviewer for the constructive comment. The rising charging current results from a bigger active surface. The sentence “Compared with SPCE, p-SPCE shows a significantly increased charging current (Inset in Figure 2a), indicating an increase in the electroactive area of the electrode through electrochemical polarization.” is added in the revision (See section 3.1. Electrochemical Polarization of SPCE).

Comment 4:The silanol group pKa 2-3 is relatively low. Please provide a reference.

Answer: We thank the reviewer for the helpful comment. The related reference (Ref. 35) has been added in the revision.

Comment 5:95 % of Clozapine binds to the blood plasma proteins. (According to Ereshefsky L, Watanabe MD, Tran-Johnson TK. Clozapine: an atypical antipsychotic agent. Clin Pharm 1989;8:691-709). This could pose a problem for very strict (e.g., FDA) verification procedures.

Answer: We thank the reviewer for the comment. The pointed work is very useful to help us further understand pharmacokinetics of CLZ. Thus, this highly related work has been added as a reference of the revision (Ref. 4). As described in the pharmacokinetics section of this paper, up to 95% of an oral dose of clozapine is detected in the blood after 3.5 hours.

Comment 6:Please provide curves at the concentration levels around the LOD. The lowest presented concentrations are at concentrations two orders higher.

Answer: We thank the reviewer for the insightful comment. As suggested, curves at the concentration levels around the LOD are shown in new figure 5a. In addition, the concentration of the detected CLZ for each curve has been added in the figure caption. The enlarged image of curves at the low concentration has also been shown as inset figure (See new figure 5a).

Comment 7:The Language is simple, descriptive, and understandable with minimum grammar errors."To improve the sensitivity for the detecting CLZ, the determination conditions such as pH, enrichment time, and ionic strength are optimized." - use past tense (were optimized).

Answer: We thank the reviewer for the careful reading and helpful comments. Accordingly, the pointed errors have been corrected in the revision.

Comment 8:Several terms can be improved to convey the correct meaning. E.g.: detection (analyte present/not present) vs. determination (what is the concentration of the analyte) Measurement vs. experiment.

Answer: We thank the reviewer for the helpful comment. As suggested, the pointed term have been improved.

Comment 9:Measurement vs. experiment.

Answer: We thank the reviewer for the helpful comment. As suggested, the title of section has been changed to “2.2. Experiments and Instrumentations”.

Comment 10:"fouling" generally means getting a decreased signal after repeated experiments at a single electrode, typically by covering the electrode with polymers etc. Anti-fouling properties in this sense are not really necessary for a disposable sensor.

Answer: We thank the reviewer for the helpful comment. The regeneration and reusability of the developed sensor have been added in the revision. The results have been shown as new Figure 7 c and d. In addition, the related discussion has also been added in the revision (See section 3.6. Detection of CLZ in Human Whole Blood with Low Sample Consumption). As shown in Figure 7 c, VMSF/p-SPCE sensor can be reused with no significant change in the response to CLZ. However, the peak currents for CLZ obtained on p-SPCE are significantly reduced (Figure 7d). This phenomenon proves the anti-fouling performance of VMSF.

Round 2

Reviewer 1 Report

Thanks for all the corrections, but the determination of CLZ in real samples is still not correct. The authors should study what dilution is necessary to do, and why it is necessary to dilute the sample. To carry out a correct study, the sample is first contaminated and then diluted, not the other way around as it is done now.

Author Response

Comment 1:“Thanks for all the corrections, but the determination of CLZ in real samples is still not correct. The authors should study what dilution is necessary to do, and why it is necessary to dilute the sample. To carry out a correct study, the sample is first contaminated and then diluted, not the other way around as it is done now.”

Answer: We thank the reviewer for the careful reading and helpful comments. In this work, PBS (0.1 M, pH=6) was used as the electrolyte for the detection. Thus, it is necessary to dilute the real sample to adjust the pH value. As suggested by the reviewer, the sample is firstly contaminated and then diluted for the analysis of real sample. The result is given as new Table 1 in the revision. The corresponding experimental details and discussion have been corrected in the revision (See sections “2.5. Electrochemical Determination of CLZ” and 3.6. “Determination of CLZ in Human Whole Blood with Low Sample Consumption”). We hope that the reviewer would now find this revision acceptable for publication.